# Reinforcement Learning with Convex Constraints

**Sobhan Miryoosefi**
Princeton University
miryoosefi@cs.princeton.edu

**Kianté Brantley**
University of Maryland
kdbrant@cs.umd.edu

**Hal Daumé III**
Microsoft Research
University of Maryland
me@hal3.name

**Miroslav Dudík**
Microsoft Research
mdudik@microsoft.com

**Robert E. Schapire**
Microsoft Research
schapire@microsoft.com

## Abstract

In standard reinforcement learning (RL), a learning agent seeks to optimize the overall reward. However, many key aspects of a desired behavior are more naturally expressed as constraints. For instance, the designer may want to limit the use of unsafe actions, increase the diversity of trajectories to enable exploration, or approximate expert trajectories when rewards are sparse. In this paper, we propose an algorithmic scheme that can handle a wide class of constraints in RL tasks, specifically, any constraints that require expected values of some vector measurements (such as the use of an action) to lie in a convex set. This captures previously studied constraints (such as safety and proximity to an expert), but also enables new classes of constraints (such as diversity). Our approach comes with rigorous theoretical guarantees and only relies on the ability to approximately solve standard RL tasks. As a result, it can be easily adapted to work with any model-free or model-based RL algorithm. In our experiments, we show that it matches previous algorithms that enforce safety via constraints, but can also enforce new properties that these algorithms cannot incorporate, such as diversity.

## 1   Introduction

Reinforcement learning (RL) typically considers the problem of learning to optimize the behavior of an agent in an unknown environment against a single scalar reward function. For simple tasks, this can be sufficient, but for complex tasks, boiling down the learning goal into a single scalar reward can be challenging. Moreover, a scalar reward might not be a natural formalism for stating certain learning objectives, such as safety desires ("avoid dangerous situations") or exploration suggestions ("maintain a distribution over visited states that is as close to uniform as possible"). In these settings, it is much more natural to define the learning goal in terms of a vector of *measurements* over the behavior of the agent, and to learn a policy whose measurement vector is inside a target set (§2).

We derive an algorithm, *approachability-based policy optimization* (APPROPO, pronounced like "apropos"), for solving such problems (§3). Given a Markov decision process with vector-valued measurements (§2), and a target constraint set, APPROPO learns a stochastic policy whose expected measurements fall in that target set (akin to Blackwell approachability in single-turn games, Blackwell, 1956). We derive our algorithm from a game-theoretic perspective, leveraging recent results in online convex optimization. APPROPO is implemented as a *reduction* to any off-the-shelf reinforcement learning algorithm that can return an approximately optimal policy, and so can be used in conjunction with the algorithms that are the most appropriate for any given domain.

Our approach builds on prior work for reinforcement learning under constraints, such as the formalism of constrained Markov decision processes (CMDPs) introduced by Altman (1999). In CMDPs, the agent's goal is to maximize reward while satisfying some linear constraints over auxiliary costs (akin to our measurements). Altman (1999) gave an LP-based approach when the CMDP is fully known, and more recently, model-free approaches have been developed for CMDPs in high-dimensional settings. For instance, Achiam et al.'s (2017) constrained policy optimization (CPO) focuses on safe exploration and seeks to ensure approximate constraint satisfaction during the learning process. Tessler et al.'s (2019) reward constrained policy optimization (RCPO) follows a two-timescale primal-dual approach, giving guarantees for the convergence to a fixed point. Le et al. (2019) describe a batch off-policy algorithm with PAC-style guarantees for CMDPs using a similar game-theoretic formulation to ours.

While all of these works are only applicable to *orthant* constraints, our algorithm can work with arbitrary convex constraints. This enables APPROPO to incorporate previously studied constraint types, such as inequality constraints that represent safety or that keep the policy's behavior close to that of an expert (Syed and Schapire, 2008), as well as constraints like the aforementioned exploration suggestion, implemented as an entropy constraint on the policy's state visitation vector. The entropy of the visitation vector was recently studied as the objective by Hazan et al. (2018), who gave an algorithm capable of maximizing a concave function (e.g., entropy) over such vectors. However, it is not clear whether their approach can be adapted to the convex constraints setting studied here.

Our main contributions are: (1) a new algorithm, APPROPO, for solving reinforcement learning problems with arbitrary convex constraints; (2) a rigorous theoretical analysis that demonstrates that it can achieve sublinear regret under mild assumptions (§3); and (3) a preliminary experimental comparison with RCPO (Tessler et al., 2019), showing that our algorithm is competitive with RCPO on orthant constraints, while also handling a diversity constraint (§4).

## 2    Setup and preliminaries: Defining the feasibility problem

We begin with a description of our learning setting. A *vector-valued Markov decision process* is a tuple $M = (\mathcal{S}, \mathcal{A}, \beta, P_s, P_z)$, where $\mathcal{S}$ is the set of states, $\mathcal{A}$ is the set of actions and $\beta$ is the initial-state distribution. Each episode starts by drawing an initial state $s_0$ from the distribution $\beta$. Then in each step $i = 1, 2, \ldots$, the agent observes its current state $s_i$ and takes action $a_i \in \mathcal{A}$ causing the environment to move to the next state $s_{i+1} \sim P_s(\cdot | s_i, a_i)$. The episode ends after a certain number of steps (called the horizon) or when a terminal state is reached. However, in our setting, instead of receiving a scalar reward, the agent observes a $d$-dimensional *measurement* vector $\mathbf{z}_i \in \mathbb{R}^d$, which, like $s_{i+1}$, is dependent on both the current state $s_i$ and the action $a_i$, that is, $\mathbf{z}_i \sim P_z(\cdot | s_i, a_i)$. (Although not explicit in our setting, reward could be incorporated in the measurement vector.)

Typically, actions are selected according to a (stationary) policy $\pi$ so that $a_i \sim \pi(s_i)$, where $\pi$ maps states to distributions over actions. We assume we are working with policies from some candidate space $\Pi$. For simplicity of presentation, we assume this space is finite, though possibly extremely large. For instance, if $\mathcal{S}$ and $\mathcal{A}$ are finite, then $\Pi$ might consist of all deterministic policies. (Our results hold also when $\Pi$ is infinite with minor technical adjustments.)

Our aim is to control the MDP so that measurements satisfy some constraints. For any policy $\pi$, we define the *long-term measurement* $\overline{\mathbf{z}}(\pi)$ as the expected sum of discounted measurements:

$$\overline{\mathbf{z}}(\pi) \triangleq \mathbb{E}\left[\sum_{i=0}^{\infty} \gamma^i \mathbf{z}_i \ \middle| \ \pi\right] \tag{1}$$

for some discount factor $\gamma \in [0, 1)$, and where expectation is over the random process described above (including randomness inherent in $\pi$).

Later, we will also find it useful to consider *mixed policies* $\mu$, which are distributions over finitely many stationary policies. The space of all such mixed policies over $\Pi$ is denoted $\Delta(\Pi)$. To execute a mixed policy $\mu$, before taking any actions, a single policy $\pi$ is randomly selected according to $\mu$; then all actions henceforth are chosen from $\pi$, for the entire episode. The long-term measurement of a mixed policy $\overline{\mathbf{z}}(\mu)$ is defined accordingly:

$$\overline{\mathbf{z}}(\mu) \triangleq \mathbb{E}_{\pi \sim \mu}\left[\overline{\mathbf{z}}(\pi)\right] = \sum_{\pi} \mu(\pi) \overline{\mathbf{z}}(\pi). \tag{2}$$

Our learning problem, called the *feasibility problem*, is specified by a convex *target set* $\mathcal{C}$. The goal is to find a mixed policy $\mu$ whose long-term measurements lie in the set $\mathcal{C}$:

$$\text{Feasibility Problem:} \quad \text{Find } \mu \in \Delta(\Pi) \text{ such that } \overline{\mathbf{z}}(\mu) \in \mathcal{C}. \tag{3}$$

For instance, in our experiments (§4) we consider a grid-world environment where the measurements include the distance traveled, an indicator of hitting a rock, and indicators of visiting various locations on the grid. The feasibility goal is to achieve at most a certain trajectory length while keeping the probability of hitting the rock below a threshold for safety reasons, and maintaining a distribution over visited states close to the uniform distribution to enable exploration. We can potentially also handle settings where the goal is to maximize one measurement (e.g., "reward") subject to others by performing a binary search over the maximum attainable value of the reward (see §3.4).

## 3   Approach, algorithm, and analysis

Before giving details of our approach, we overview the main ideas, which, to a large degree, follow the work of Abernethy et al. (2011), who considered the problem of solving two-player games; we extend these results to solve our feasibility problem (3).

Although feasibility is our main focus, we actually solve the stronger problem of finding a mixed policy $\mu$ that minimizes the Euclidean distance between $\overline{\mathbf{z}}(\mu)$ and $\mathcal{C}$, meaning the Euclidean distance between $\overline{\mathbf{z}}(\mu)$ and its closest point in $\mathcal{C}$. That is, we want to solve

$$\min_{\mu \in \Delta(\Pi)} \text{dist}(\overline{\mathbf{z}}(\mu), \mathcal{C}) \tag{4}$$

where $\text{dist}$ denotes the Euclidean distance between a point and a set.

Our main idea is to take a game-theoretic approach, formulating this problem as a game and solving it. Specifically, suppose we can express the distance function in Eq. (4) as a maximization of the form

$$\text{dist}(\overline{\mathbf{z}}(\mu), \mathcal{C}) = \max_{\boldsymbol{\lambda} \in \Lambda} \boldsymbol{\lambda} \cdot \overline{\mathbf{z}}(\mu) \tag{5}$$

for some convex, compact set $\Lambda$.[1] Then Eq. (4) becomes

$$\min_{\mu \in \Delta(\Pi)} \max_{\boldsymbol{\lambda} \in \Lambda} \boldsymbol{\lambda} \cdot \overline{\mathbf{z}}(\mu). \tag{6}$$

This min-max form immediately evokes interpretation as a two-person zero-sum game: the first player chooses a mixed policy $\mu$, the second player responds with a vector $\boldsymbol{\lambda}$, and $\boldsymbol{\lambda} \cdot \overline{\mathbf{z}}(\mu)$ is the amount that the first player is then required to pay to the second player. Assuming this game satisfies certain conditions, the final payout under the optimal play, called the *value* of the game, is the same even when the order of the players is reversed:

$$\max_{\boldsymbol{\lambda} \in \Lambda} \min_{\mu \in \Delta(\Pi)} \boldsymbol{\lambda} \cdot \overline{\mathbf{z}}(\mu). \tag{7}$$

Note that the policy $\mu$ we are seeking is the *solution* of this game, that is, the policy realizing the minimum in Eq. (6). Therefore, to find that policy, we can apply general techniques for solving a game, namely, to let a no-regret learning algorithm play the game repeatedly against a best-response player. When played in this way, it can be shown that the averages of their plays converge to the solution of the game (details in §3.1).

In our case, we can use a no-regret algorithm for the $\boldsymbol{\lambda}$-player, and best response for the $\mu$-player. Importantly, in our context, computing best response turns out to be an especially convenient task. Given $\boldsymbol{\lambda}$, best response means finding the mixed policy $\mu$ minimizing $\boldsymbol{\lambda} \cdot \overline{\mathbf{z}}(\mu)$. As we show below, this can be solved by treating the problem as a standard reinforcement learning task where in each step $i$, the agent accrues a scalar reward $r_i = -\boldsymbol{\lambda} \cdot \mathbf{z}_i$. We refer to any algorithm for solving the problem of scalar reward maximization as the *best-response oracle*. During the run of our algorithm, we invoke this oracle for different vectors $\boldsymbol{\lambda}$ corresponding to different definitions of a scalar reward.

**Algorithm 1** Solving a game with repeated play
---
1: **input** concave-convex function $g : \Lambda \times \mathcal{U} \to \mathbb{R}$, online learning algorithm LEARNER
2: **for** $t = 1$ **to** $T$ **do**
3:     LEARNER makes a decision $\boldsymbol{\lambda}_t \in \Lambda$
4:     $\mathbf{u}_t \leftarrow \operatorname{argmin}_{\mathbf{u} \in \mathcal{U}} g(\boldsymbol{\lambda}_t, \mathbf{u})$
5:     LEARNER observes loss function $\ell_t(\boldsymbol{\lambda}) = -g(\boldsymbol{\lambda}, \mathbf{u}_t)$
6: **end for**
7: **return** $\overline{\boldsymbol{\lambda}} = \frac{1}{T} \sum_{t=1}^{T} \boldsymbol{\lambda}_t$ and $\overline{\mathbf{u}} = \frac{1}{T} \sum_{t=1}^{T} \mathbf{u}_t$
---

Although the oracle is only capable of solving RL tasks with a scalar reward, our algorithm can leverage this capability to solve the multi-dimensional feasibility (or distance minimization) problem.

In the remainder of this section, we provide the details of our approach, leading to our main algorithm and its analysis, and conclude with a discussion of steps for making a practical implementation. We begin by discussing game-playing techniques in general, which we then apply to our setting.

## 3.1 Solving zero-sum games using online learning

At the core of our approach, we use the general technique of Freund and Schapire (1999) for solving a game by repeatedly playing a no-regret online learning algorithm against best response.

For this purpose, we first briefly review the framework of online convex optimization, which we will soon use for one of the players: At time $t = 1, \ldots, T$, the learner makes a decision $\boldsymbol{\lambda}_t \in \Lambda$, the environment reveals a convex loss function $\ell_t : \Lambda \to \mathbb{R}$, and the learner incurs loss $\ell_t(\boldsymbol{\lambda}_t)$. The learner seeks to achieve small *regret*, the gap between its loss and the best in hindsight:

$$\text{Regret}_T \triangleq \left[ \sum_{t=1}^{T} \ell_t(\boldsymbol{\lambda}_t) \right] - \min_{\boldsymbol{\lambda} \in \Lambda} \left[ \sum_{t=1}^{T} \ell_t(\boldsymbol{\lambda}) \right]. \tag{8}$$

An online learning algorithm is *no-regret* if $\text{Regret}_T = o(T)$, meaning its average loss approaches the best in hindsight. An example of such an algorithm is *online gradient descent (OGD)* of Zinkevich (2003) (see Appendix A). If the Euclidean diameter of $\Lambda$ is at most $D$, and $\|\nabla \ell_t(\boldsymbol{\lambda})\| \leq G$ for any $t$ and $\boldsymbol{\lambda} \in \Lambda$, then the regret of OGD is at most $DG\sqrt{T}$.

Now consider a two-player zero-sum game in which two players select, respectively, $\boldsymbol{\lambda} \in \Lambda$ and $\mathbf{u} \in \mathcal{U}$, resulting in a payout of $g(\boldsymbol{\lambda}, \mathbf{u})$ from the $\mathbf{u}$-player to the $\boldsymbol{\lambda}$-player. The $\boldsymbol{\lambda}$-player wants to maximize this quantity and the $\mathbf{u}$-player wants to minimize it. Assuming $g$ is concave in $\boldsymbol{\lambda}$ and convex in $\mathbf{u}$, if both spaces $\Lambda$ and $\mathcal{U}$ are convex and compact, then the minimax theorem (von Neumann, 1928; Sion, 1958) implies that

$$\max_{\boldsymbol{\lambda} \in \Lambda} \min_{\mathbf{u} \in \mathcal{U}} g(\boldsymbol{\lambda}, \mathbf{u}) = \min_{\mathbf{u} \in \mathcal{U}} \max_{\boldsymbol{\lambda} \in \Lambda} g(\boldsymbol{\lambda}, \mathbf{u}). \tag{9}$$

This means that the $\boldsymbol{\lambda}$-player has an "optimal" strategy which realizes the maximum on the left and guarantees payoff of at least the *value* of the game, i.e., the value given by this expression; a similar statement holds for the $\mathbf{u}$-player.

We can solve this game (find these optimal strategies) by playing it repeatedly. We use a no-regret online learner as the $\boldsymbol{\lambda}$-player. At each time $t = 1, \ldots, T$, the learner chooses $\boldsymbol{\lambda}_t \in \Lambda$. In response, the $\mathbf{u}$-player, who in this setting is permitted knowledge of $\boldsymbol{\lambda}_t$, selects $\mathbf{u}_t$ to minimize the payout, that is, $\mathbf{u}_t = \operatorname{argmin}_{\mathbf{u} \in \mathcal{U}} g(\boldsymbol{\lambda}_t, \mathbf{u})$. This is called *best response*. The online learning algorithm is then updated by setting its loss function to be $\ell_t(\boldsymbol{\lambda}) = -g(\boldsymbol{\lambda}, \mathbf{u}_t)$. (See Algorithm 1.) As stated in Theorem 3.1, $\overline{\boldsymbol{\lambda}}$ and $\overline{\mathbf{u}}$, the averages of the players' decisions, converge to the solution of the game (see Appendix B for the proof).

**Theorem 3.1.** *Let $v$ be the value of the game in Eq. (9) and let $\text{Regret}_T$ be the regret of the $\boldsymbol{\lambda}$-player. Then for $\overline{\boldsymbol{\lambda}}$ and $\overline{\mathbf{u}}$ we have*

$$\min_{\mathbf{u} \in \mathcal{U}} g(\overline{\boldsymbol{\lambda}}, \mathbf{u}) \geq v - \delta \quad and \quad \max_{\boldsymbol{\lambda} \in \Lambda} g(\boldsymbol{\lambda}, \overline{\mathbf{u}}) \leq v + \delta, \quad where \; \delta = \frac{1}{T} \text{Regret}_T. \tag{10}$$

## 3.2 Algorithm and main result

We can now apply this game-playing framework to the approach outlined at the beginning of this section. First, we show how to write distance as a maximization, as in Eq. (5). For now, we assume that our target set $\mathcal{C}$ is a *convex cone*, that is, closed under summation and also multiplication by non-negative scalars (we will remove this assumption in §3.3). With this assumption, we can apply the following lemma (Lemma 13 of Abernethy et al., 2011), in which distance to a convex cone $\mathcal{C} \subseteq \mathbb{R}^d$ is written as a maximization over a dual convex cone $\mathcal{C}^\circ$ called the *polar cone*:

$$\mathcal{C}^\circ \triangleq \{\boldsymbol{\lambda} : \; \boldsymbol{\lambda} \cdot \mathbf{x} \leq 0 \text{ for all } \mathbf{x} \in \mathcal{C}\}. \tag{11}$$

**Lemma 3.2.** *For a convex cone $\mathcal{C} \subseteq \mathbb{R}^d$ and any point $\mathbf{x} \in \mathbb{R}^d$*

$$\text{dist}(\mathbf{x}, \mathcal{C}) = \max_{\boldsymbol{\lambda} \in \mathcal{C}^\circ \cap \mathcal{B}} \boldsymbol{\lambda} \cdot \mathbf{x}, \tag{12}$$

*where $\mathcal{B} \triangleq \{\mathbf{x} : \; \|\mathbf{x}\| \leq 1\}$ is the Euclidean ball of radius 1 at the origin.*

Thus, Eq. (5) is immediately achieved by setting $\Lambda = \mathcal{C}^\circ \cap \mathcal{B}$, so the distance minimization problem (4) can be cast as the min-max problem (6). This is a special case of the zero-sum game (9), with $\mathcal{U} = \{\bar{\mathbf{z}}(\mu) : \; \mu \in \Delta(\Pi)\}$ and $g(\boldsymbol{\lambda}, \mathbf{u}) = \boldsymbol{\lambda} \cdot \mathbf{u}$, which can be solved with Algorithm 1. Note that the set $\mathcal{U}$ is convex and compact, because it is a linear transformation of a convex and compact set $\Delta(\Pi)$.

We will see below that the best responses $\mathbf{u}_t$ in Algorithm 1 can be expressed as $\bar{\mathbf{z}}(\pi_t)$ for some $\pi_t \in \Pi$, and so Algorithm 1 returns

$$\bar{\mathbf{u}} = \frac{1}{T} \sum_{t=1}^{T} \bar{\mathbf{z}}(\pi_t) = \bar{\mathbf{z}}\left(\frac{1}{T} \sum_{t=1}^{T} \pi_t\right),$$

which is exactly the long-term measurement vector of the mixed policy $\bar{\mu} = \frac{1}{T} \sum_{t=1}^{T} \pi_t$. For this mixed policy, Theorem 3.1 immediately implies

$$\text{dist}(\bar{\mathbf{z}}(\bar{\mu}), \mathcal{C}) \leq \min_{\mu \in \Delta(\Pi)} \text{dist}(\bar{\mathbf{z}}(\mu), \mathcal{C}) + \tfrac{1}{T} \text{Regret}_T. \tag{13}$$

If the problem is feasible, then $\min_{\mu \in \Delta(\Pi)} \text{dist}(\bar{\mathbf{z}}(\mu), \mathcal{C}) = 0$, and since $\text{Regret}_T = o(T)$, our long-term measurement $\bar{\mathbf{z}}(\bar{\mu})$ converges to the target set and solves the feasibility problem (3). It remains to specify how to implement the no-regret learner for the $\boldsymbol{\lambda}$-player and best response for the $\mathbf{u}$-player. We discuss these next, beginning with the latter.

The best-response player, for a given $\boldsymbol{\lambda}$, aims to minimize $\boldsymbol{\lambda} \cdot \bar{\mathbf{z}}(\mu)$ over mixed policies $\mu$, but since this objective is linear in the mixture weights $\mu(\pi)$ (see Eq. 2), it suffices to minimize $\boldsymbol{\lambda} \cdot \bar{\mathbf{z}}(\pi)$ over stationary policies $\pi \in \Pi$. The key point, as already mentioned, is that this is the same as finding a policy that maximizes long-term reward in a standard reinforcement learning task if we define the scalar reward to be $r_i = -\boldsymbol{\lambda} \cdot \mathbf{z}_i$. This is because the reward of a policy $\pi$ is given by

$$R(\pi) \triangleq \mathbb{E}\left[\sum_{i=0}^{\infty} \gamma^i r_i \; \Big| \; \pi\right] = \mathbb{E}\left[\sum_{i=0}^{\infty} \gamma^i(-\boldsymbol{\lambda} \cdot \mathbf{z}_i) \; \Big| \; \pi\right] = -\boldsymbol{\lambda} \cdot \mathbb{E}\left[\sum_{i=0}^{\infty} \gamma^i \mathbf{z}_i \; \Big| \; \pi\right] = -\boldsymbol{\lambda} \cdot \bar{\mathbf{z}}(\pi). \tag{14}$$

Therefore, maximizing $R(\pi)$, as in standard RL, is equivalent to minimizing $\boldsymbol{\lambda} \cdot \bar{\mathbf{z}}(\pi)$.

Thus, best response can be implemented using any one of the many well-studied RL algorithms that maximize a scalar reward. We refer to such an RL algorithm as the *best-response oracle*. For robustness, we allow this oracle to return an approximately optimal policy.

**Best-response oracle:** BESTRESPONSE($\boldsymbol{\lambda}$).
Given $\boldsymbol{\lambda} \in \mathbb{R}^d$, return a policy $\pi \in \Pi$ that satisfies $R(\pi) \geq \max_{\pi' \in \Pi} R(\pi') - \epsilon_0$, where $R(\pi)$ is the long-term reward of policy $\pi$ with scalar reward defined as $r = -\boldsymbol{\lambda} \cdot \mathbf{z}$.

For the $\boldsymbol{\lambda}$-player, we do our analysis using online gradient descent (Zinkevich, 2003), an effective no-regret learner. For its update, OGD needs the gradient of the loss functions $\ell_t(\boldsymbol{\lambda}) = -\boldsymbol{\lambda} \cdot \bar{\mathbf{z}}(\pi_t)$, which is just $-\bar{\mathbf{z}}(\pi_t)$. With access to the MDP, $\bar{\mathbf{z}}(\pi)$ can be estimated simply by generating multiple trajectories using $\pi$ and averaging the observed measurements. We formalize this by assuming access to an *estimation oracle* for estimating $\bar{\mathbf{z}}(\pi)$.

---
**Algorithm 2** APPROPO
---
1: **input** projection oracle $\Gamma_{\mathcal{C}}(\cdot)$ for target set $\mathcal{C}$ which is a convex cone,
        best-response oracle BESTRESPONSE$(\cdot)$, estimation oracle EST$(\cdot)$,
        step size $\eta$, number of iterations $T$
2: **define** $\Lambda \triangleq \mathcal{C}^\circ \cap \mathcal{B}$, and its projection operator $\Gamma_\Lambda(\mathbf{x}) \triangleq (\mathbf{x} - \Gamma_{\mathcal{C}}(\mathbf{x}))/\max\{1, \|\mathbf{x} - \Gamma_{\mathcal{C}}(\mathbf{x})\|\}$
3: **initialize** $\boldsymbol{\lambda}_1$ arbitrarily in $\Lambda$
4: **for** $t = 1$ **to** $T$ **do**
5:     Compute an approximately optimal policy for standard RL with scalar reward $r = -\boldsymbol{\lambda}_t \cdot \mathbf{z}$:
        $\pi_t \leftarrow$ BESTRESPONSE$(\boldsymbol{\lambda}_t)$
6:     Call the estimation oracle to approximate long-term measurement for $\pi_t$:
        $\hat{\mathbf{z}}_t \leftarrow$ EST$(\pi_t)$
7:     Update $\boldsymbol{\lambda}_t$ using online gradient descent with the loss function $\ell_t(\boldsymbol{\lambda}) = -\boldsymbol{\lambda} \cdot \hat{\mathbf{z}}_t$:
        $\boldsymbol{\lambda}_{t+1} \leftarrow \Gamma_\Lambda\big(\boldsymbol{\lambda}_t + \eta\hat{\mathbf{z}}_t\big)$
8: **end for**
9: **return** $\bar{\mu}$, a uniform mixture over $\pi_1, \ldots, \pi_T$
---

**Estimation oracle:** EST$(\pi)$.
    Given policy $\pi$, return $\hat{\mathbf{z}}$ satisfying $\|\hat{\mathbf{z}} - \bar{\mathbf{z}}(\pi)\| \leq \epsilon_1$.

OGD also requires projection to the set $\Lambda = \mathcal{C}^\circ \cap \mathcal{B}$. In fact, if we can simply project onto the target set $\mathcal{C}$, which is more natural, then it is possible to also project onto $\Lambda$. Consider an arbitrary $\mathbf{x}$ and denote its projection onto $\mathcal{C}$ as $\Gamma_{\mathcal{C}}(\mathbf{x})$. Then the projection of $\mathbf{x}$ onto the polar cone $\mathcal{C}^\circ$ is $\Gamma_{\mathcal{C}^\circ}(\mathbf{x}) = \mathbf{x} - \Gamma_{\mathcal{C}}(\mathbf{x})$ (Ingram and Marsh, 1991). Given the projection $\Gamma_{\mathcal{C}^\circ}(\mathbf{x})$ and further projecting onto $\mathcal{B}$, we obtain $\Gamma_\Lambda(\mathbf{x}) = (\mathbf{x} - \Gamma_{\mathcal{C}}(\mathbf{x}))/\max\{1, \|\mathbf{x} - \Gamma_{\mathcal{C}}(\mathbf{x})\|\}$ (because Dykstra's projection algorithm converges to this point after two steps, Boyle and Dykstra, 1986). Therefore, it suffices to require access to a *projection oracle* for $\mathcal{C}$:

**Projection oracle:** $\Gamma_{\mathcal{C}}(\mathbf{x}) = \text{argmin}_{\mathbf{x}' \in \mathcal{C}} \|\mathbf{x} - \mathbf{x}'\|$.

Pulling these ideas together and plugging into Algorithm 1, we obtain our main algorithm, called APPROPO (Algorithm 2), for *approachability-based policy optimization*. The algorithm provably yields a mixed policy that approximately minimizes distance to the set $\mathcal{C}$, as shown in Theorem 3.3 (proved in Appendix C).

**Theorem 3.3.** *Assume that $\mathcal{C}$ is a convex cone and for all measurements we have $\|\mathbf{z}\| \leq B$. Suppose we run Algorithm 2 for $T$ rounds with $\eta = \big(\frac{B}{1-\gamma} + \epsilon_1\big)^{-1}T^{-1/2}$. Then*

$$\text{dist}(\bar{\mathbf{z}}(\bar{\mu}), \mathcal{C}) \leq \min_{\mu \in \Delta(\Pi)} \text{dist}(\bar{\mathbf{z}}(\mu), \mathcal{C}) + \big(\tfrac{B}{1-\gamma} + \epsilon_1\big)T^{-1/2} + \epsilon_0 + 2\epsilon_1, \qquad (15)$$

*where $\bar{\mu}$ is the mixed policy returned by the algorithm.*

When the goal is to solve the feasibility problem (3) rather than the stronger distance minimization (4), we can make use of a weaker reinforcement learning oracle, which only needs to find a policy that is "good enough" in the sense of providing long-term reward above some threshold:

**Positive-response oracle:** POSRESPONSE$(\boldsymbol{\lambda})$.
    Given $\boldsymbol{\lambda} \in \mathbb{R}^d$, return $\pi \in \Pi$ that satisfies $R(\pi) \geq -\epsilon_0$ if $\max_{\pi' \in \Pi} R(\pi') \geq 0$ (and arbitrary $\pi$ otherwise), where $R(\pi)$ is the long-term reward of $\pi$ with scalar reward $r = -\boldsymbol{\lambda} \cdot \mathbf{z}$.

When the problem is feasible, it can be shown that there must exist $\pi \in \Pi$ with $R(\pi) \geq 0$, and furthermore, that $\ell_t(\boldsymbol{\lambda}_t) \geq -(\epsilon_0 + \epsilon_1)$ (from Lemma C.1 in Appendix C). This means, if the goal is feasibility, we can modify Algorithm 2, replacing BESTRESPONSE with POSRESPONSE, and adding a test at the end of each iteration to report infeasibility if $\ell_t(\boldsymbol{\lambda}_t) < -(\epsilon_0 + \epsilon_1)$. The pseudocode is provided in Algorithm 4 in Appendix D along with the proof of the following convergence bound:

**Theorem 3.4.** *Assume that $\mathcal{C}$ is a convex cone and for all measurements we have $\|\mathbf{z}\| \leq B$. Suppose we run Algorithm 4 for $T$ rounds with $\eta = \big(\frac{B}{1-\gamma} + \epsilon_1\big)^{-1}T^{-1/2}$. Then either the algorithm reports infeasibility or returns $\bar{\mu}$ such that*

$$\text{dist}(\bar{\mathbf{z}}(\bar{\mu}), \mathcal{C}) \leq \big(\tfrac{B}{1-\gamma} + \epsilon_1\big)T^{-1/2} + \epsilon_0 + 2\epsilon_1. \qquad (16)$$

### 3.3 Removing the cone assumption

Our results so far have assumed the target set $\mathcal{C}$ is a convex cone. If instead $\mathcal{C}$ is an arbitrary convex, compact set, we can use the technique of Abernethy et al. (2011) and apply our algorithm to a specific convex cone $\tilde{\mathcal{C}}$ constructed from $\mathcal{C}$ to obtain a solution with provable guarantees.

In more detail, given a compact, convex target set $\mathcal{C} \subseteq \mathbb{R}^d$, we augment every vector in $\mathcal{C}$ with a new coordinate held fixed to some value $\kappa > 0$, and then let $\tilde{\mathcal{C}}$ be its conic hull. Thus,

$$\tilde{\mathcal{C}} = \text{cone}(\mathcal{C} \times \{\kappa\}), \qquad \text{where } \text{cone}(\mathcal{X}) = \{\alpha\mathbf{x} \mid \mathbf{x} \in \mathcal{X}, \alpha \geq 0\}. \tag{17}$$

Given our original vector-valued MDP $M = (\mathcal{S}, \mathcal{A}, \beta, P_s, P_z)$, we define a new MDP $M' = (\mathcal{S}, \mathcal{A}, \beta, P_s, P'_{z'})$ with $(d+1)$-dimensional measurement $\mathbf{z}' \in \mathbb{R}^{d+1}$, defined (and generated) by

$$\mathbf{z}'_i = \mathbf{z}_i \oplus \langle (1-\gamma)\kappa \rangle, \qquad \mathbf{z}_i \sim P_z(\cdot \mid s_i, a_i) \tag{18}$$

where $\oplus$ denotes vector concatenation. Writing long-term measurement for $M$ and $M'$ as $\bar{\mathbf{z}}$ and $\bar{\mathbf{z}}'$ respectively, $\bar{\mathbf{z}}'(\pi) = \bar{\mathbf{z}}(\pi) \oplus \langle \kappa \rangle$, for any policy $\pi \in \Pi$, and similarly for any mixed policy $\mu$.

The main idea is to apply the algorithms described above to the modified MDP $M'$ using the cone $\tilde{\mathcal{C}}$ as target set. For an appropriate choice of $\kappa > 0$, we show that the resulting mixed policy will approximately minimize distance to $\mathcal{C}$ for the original MDP $M$. This is a consequence of the following lemma, an extension of Lemma 14 of Abernethy et al. (2011), which shows that distances are largely preserved in a controllable way under this construction. The proof is in Appendix E.

**Lemma 3.5.** *Consider a compact, convex set $\mathcal{C}$ in $\mathbb{R}^d$ and $\mathbf{x} \in \mathbb{R}^d$. For any $\delta > 0$, let $\tilde{\mathcal{C}} = \text{cone}(\mathcal{C} \times \{\kappa\})$, where $\kappa = \frac{\max_{\mathbf{x} \in \mathcal{C}} \|\mathbf{x}\|}{\sqrt{2\delta}}$. Then $\text{dist}(\mathbf{x}, \mathcal{C}) \leq (1+\delta)\text{dist}(\mathbf{x} \oplus \langle \kappa \rangle, \tilde{\mathcal{C}})$.*

**Corollary 3.6.** *Assume that $\mathcal{C}$ is a convex, compact set and for all measurements we have $\|\mathbf{z}\| \leq B$. Then by putting $\eta = \left(\frac{B+\kappa}{1-\gamma} + \epsilon_1\right)^{-1} T^{-1/2}$ and running Algorithm 2 for $T$ rounds with $M'$ as the MDP and $\tilde{\mathcal{C}}$ as the target set, the mixed policy $\bar{\mu}$ returned by the algorithm satisfies*

$$\text{dist}(\bar{\mathbf{z}}(\bar{\mu}), \mathcal{C}) \leq (1+\delta)\left(\min_{\mu \in \Delta(\Pi)} \text{dist}(\bar{\mathbf{z}}(\mu), \mathcal{C}) + \left(\frac{B+\kappa}{1-\gamma} + \epsilon_1\right)T^{-1/2} + \epsilon_0 + 2\epsilon_1\right), \tag{19}$$

*where $\kappa = \frac{\max_{\mathbf{x} \in \mathcal{C}} \|\mathbf{x}\|}{\sqrt{2\delta}}$ for an arbitrary $\delta > 0$. Similarly for Algorithm 4, we either have*

$$\text{dist}(\bar{\mathbf{z}}(\bar{\mu}), \mathcal{C}) \leq (1+\delta)\left(\left(\frac{B+\kappa}{1-\gamma} + \epsilon_1\right)T^{-1/2} + \epsilon_0 + 2\epsilon_1\right) \tag{20}$$

*or the algorithm reports infeasibility.*

### 3.4 Practical implementation of the positive response and estimation oracles

We next briefly describe a few techniques for the practical implementation of our algorithm.

As discussed in §3.2, when our aim is to solve a feasibility problem, we only need access to a positive response oracle. In episodic environments, it is straightforward to use any standard iterative RL approach as a positive response oracle: As the RL algorithm runs, we track its accrued rewards, and when the trailing average of the last $n$ trajectory-level rewards goes above some level $-\epsilon$, we return the current policy (possibly specified implicitly as a $Q$-function).[2] Furthermore, the average of the measurement vectors $\mathbf{z}$ collected over the last $n$ trajectories can serve as the estimate $\hat{\mathbf{z}}_t$ of the long-term measurement required by the algorithm, side-stepping the need for an additional estimation oracle.

The hyperparameters $\epsilon$ and $n$ influence the oracle quality; specifically, assuming that the rewards are bounded and the overall number of trajectories until the oracle terminates is at most polynomial in $n$, we have $\epsilon_0 = \epsilon - O(\sqrt{(\log n)/n})$ and $\epsilon_1 = O(\sqrt{(\log n)/n})$. In principle, we could use Theorem 3.4 to select a value $T$ at which to stop; in practice, we run until the running average of the measurements $\hat{\mathbf{z}}_t$ gets within a small distance of the target set $\mathcal{C}$. If the RL algorithm runs for too long without achieving non-negative rewards, we stop and declare that the underlying problem is "empirically infeasible." (Actual infeasibility would hold if it is truly not possible to reach non-negative expected reward.)

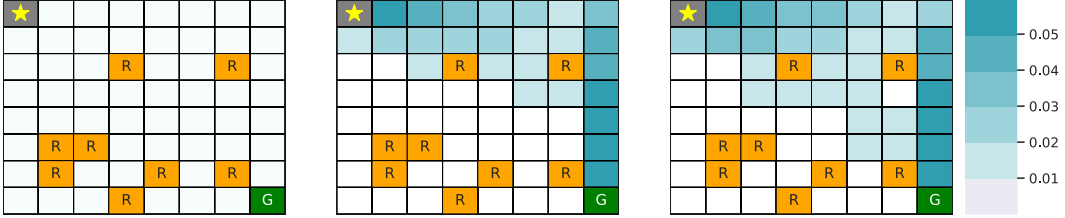

Figure 1: *Left:* The Mars rover environment. The agent starts in top-left and needs to reach the goal in bottom-right while avoiding rocks. *Middle, Right:* Visitation probabilities of APPROPO (middle) and APPROPO with a diversity constraints (right) at 12k samples. Both plots based on a single run.

An important mechanism to further speed up our algorithm is to maintain a "cache" of all the policies returned by the positive response oracle so far. Each of the cached policies $\pi$ is stored with the estimate of its expected measurement vector $\hat{\mathbf{z}}(\pi) \approx \bar{\mathbf{z}}(\pi)$, based on its last $n$ iterations (as above). In each outer-loop iteration of our algorithm, we first check if the cache contains a policy that already achieves a reward at least $-\epsilon$ under the new $\boldsymbol{\lambda}$; this can be determined from the cached $\hat{\mathbf{z}}(\pi)$ since the reward is just a linear function of the measurement vector. If such a policy is found, we return it, alongside $\hat{\mathbf{z}}(\pi)$, instead of calling the oracle. Otherwise, we pick the policy from the cache with the largest reward (below $-\epsilon$ by assumption) and use it to warm-start the RL algorithm implementing the oracle. The cache can be initialized with a few random policies (as we do in our experiments), effectively implementing randomized weight initialization.

The cache interacts well with a straightforward binary-search scheme that can be used when the goal is to maximize some reward (possibly subject to additional constraints), rather than only satisfy a set of constraints. The feasibility problems corresponding to iterates of binary search only differ in the constraint values, but use the same measurements, so the same cache can be reused across all iterations.

**Running time.** Note that APPROPO spends the bulk of its running time executing the best-response oracle. It additionally performs updates of $\boldsymbol{\lambda}$, but these tend to be orders of magnitude cheaper than any per-episode (or per-transition) updates within the oracle. For example, in our experiments (§4), the dimension of $\boldsymbol{\lambda}$ is either 2 or 66 (without or with the diversity constraint, respectively), whereas the policies $\pi$ trained by the oracle are two-layer networks described by 8,704 floating-point numbers.

## 4 Experiments

We next evaluate the performance of APPROPO and demonstrate its ability to handle a variety of constraints. For simplicity, we focus on the feasibility version (Algorithm 4 in Appendix D). We compare APPROPO with the RCPO approach of Tessler et al. (2019), which adapts policy gradient, specifically, asynchronous actor-critic (A2C) (Mnih et al., 2016), to find a fixed point of the Lagrangian of the constrained policy optimization problem. RCPO maintains and updates a vector of Lagrange multipliers, which is then used to derive a reward for A2C. The vector of Lagrange multipliers serves a similar role as our $\boldsymbol{\lambda}$, and the overall structure of RCPO is similar to APPROPO, so RCPO is a natural baseline for a comparison. Unlike APPROPO, RCPO only allows orthant constraints and it seeks to maximize reward, whereas APPROPO solves the feasibility problem.

For a fair comparison, APPROPO uses A2C as a positive-response oracle, with the same hyperparameters as used in RCPO. Online learning in the outer loop of APPROPO was implemented via online gradient descent with momentum. Both RCPO and APPROPO have an outer-loop learning rate parameter, which we tuned over a grid of values $10^{-i}$ with integer $i$ (see Appendix F for the details). Here, we report the results with the best learning rate for each method.

We ran our experiments on a small version of the *Mars rover* grid-world environment, used previously for the evaluation of RCPO (Tessler et al., 2019). In this environment, depicted in Figure 1 (left), the agent must move from the starting position to the goal without crashing into rocks. The episode terminates when a rock or the goal is reached, or after 300 steps. The environment is stochastic: with probability $\delta = 0.05$ the agent's action is perturbed to a random action. The agent receives small negative reward each time step and zero for terminating, with $\gamma = 0.99$. We used the same safety constraint as Tessler et al. (2019): ensure that the (discounted) probability of hitting a rock is at most a fixed threshold (set to 0.2). RCPO seeks to maximize reward subject to this constraint. APPROPO

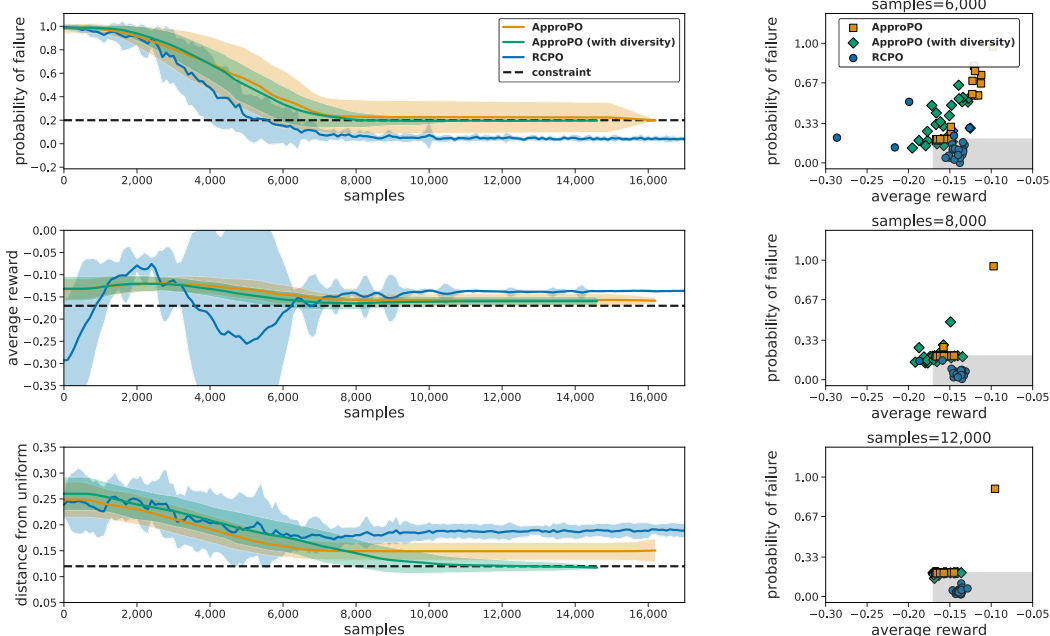

Figure 2: *Left:* The performance of the algorithms as a function of the number of samples (steps in the environment); showing average and standard deviation over 25 runs. The vertical axes correspond to the three constraints, with thresholds shown as a dashed line; for reward (middle) this is a lower bound; for the others it is an upper bound. *Right:* Each point in the scatter plot represents the reward and the probability of failure obtained by the policy learnt by the algorithm at the specified number of samples. The grey region is the target set. Different points represent different random runs.

solves a feasibility problem with the same safety constraint, and an additional constraint requiring that the reward be at least $-0.17$ (this is slightly lower than the final reward achieved by RCPO). We also experimented with including the exploration suggestion as a "diversity constraint," requiring that the Euclidean distance between our visitation probability vector (across the cells of the grid) and the uniform distribution over the upper-right triangle cells of the grid (excluding rocks) be at most $0.12$.[3]

In Figure 2 (left), we show how the probability of failure, the average reward, and the distance to the uniform distribution over upper triangle vary as a function of the number of samples seen by each algorithm. Both variants of our algorithm are able to satisfy the safety constraints and reach similar reward as RCPO with a similar number of samples (around 8k samples). Furthermore, including the diversity constraint, which RCPO is not capable of enforcing, allowed our method to reach a more diverse policy as depicted in both Figure 2 (bottom-left) and Figure 1 (right).

## 5   Conclusion

In this paper, we introduced APPROPO, an algorithm for solving reinforcement learning problems with arbitrary convex constraints. APPROPO can combine any no-regret online learner with any standard RL algorithm that optimizes a scalar reward. Theoretically, we showed that for the specific case of online gradient descent, APPROPO learns to approach the constraint set at a rate of $1/\sqrt{T}$, with an additive non-vanishing term that measures the optimality gap of the reinforcement learner. Experimentally, we demonstrated that APPROPO can be applied with well-known RL algorithms for discrete domains (like actor-critic), and achieves similar performance as RCPO (Tessler et al., 2019), while being able to satisfy additional types of constraints. In sum, this yields a theoretically justified, practical algorithm for solving the approachability problem in reinforcement learning.

## Footnotes

[1]Note that the distance between a point and a set is defined as a minimization of the distance function over all points in the set $\mathcal{C}$, but here we require that it be rewritten as a maximization of a linear function over some other set $\Lambda$. We will show how to achieve this in §3.2.

[2]This assumes that the last $n$ trajectories accurately estimate the performance of the final iterate. If that is not the case, the oracle can instead return the mixture of the policies corresponding to the last $n$ iterates.

[3]This number ensures that APPROPO without the diversity constraint does not satisfy it automatically.

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
