[Supplementary Material · paperid7857-approPO-appendix.pdf]

# Supplementary Material For:
## Reinforcement Learning with Convex Constraints

## A  Online gradient descent (OGD)

---

**Algorithm 3** Online gradient descent (OGD)

---

1: **input**: projection oracle $\Gamma_\Lambda$ $\{\Gamma_\Lambda(\boldsymbol{\lambda}) = \text{argmin}_{\boldsymbol{\lambda}' \in \Lambda} \|\boldsymbol{\lambda} - \boldsymbol{\lambda}'\|\}$
2: **init**: $\boldsymbol{\lambda}_1$ arbitrarily
3: **parameters**: step size $\eta_t$
4: **for** $t = 1$ **to** $T$ **do**
5:    observe convex loss function $\ell_t : \Lambda \to \mathbb{R}$
6:    $\boldsymbol{\lambda}'_{t+1} = \boldsymbol{\lambda}_t - \eta_t \nabla \ell_t(\boldsymbol{\lambda}_t)$
7:    $\boldsymbol{\lambda}_{t+1} = \Gamma_\Lambda(\boldsymbol{\lambda}'_{t+1})$
8: **end for**

---

**Theorem A.1.** *(Zinkevich, 2003) Assume that for any $\boldsymbol{\lambda}, \boldsymbol{\lambda}' \in \Lambda$ we have $\|\boldsymbol{\lambda} - \boldsymbol{\lambda}'\| \le D$ and also $\|\nabla \ell_t(\boldsymbol{\lambda})\| \le G$. Let $\eta_t = \eta = \frac{D}{G\sqrt{T}}$. Then the regret of OGD is*

$$\text{Regret}_T(OGD) = \sum_{t=1}^{T} \ell_t(\boldsymbol{\lambda}_t) - \min_{\boldsymbol{\lambda}} \sum_{t=1}^{T} \ell_t(\boldsymbol{\lambda}) \le DG\sqrt{T}.$$

## B  Proof of Theorem 3.1

We have that

$$\frac{1}{T}\sum_{t=1}^{T} g(\boldsymbol{\lambda}_t, \mathbf{u}_t) = \frac{1}{T}\sum_{t=1}^{T} \min_{\mathbf{u}\in\mathcal{U}} g(\boldsymbol{\lambda}_t, \mathbf{u}) \tag{21}$$

$$\le \frac{1}{T} \min_{\mathbf{u}\in\mathcal{U}} \sum_{t=1}^{T} g(\boldsymbol{\lambda}_t, \mathbf{u}) \tag{22}$$

$$\le \min_{\mathbf{u}\in\mathcal{U}} g\left(\frac{1}{T}\sum_{t=1}^{T} \boldsymbol{\lambda}_t, \mathbf{u}\right) \tag{23}$$

$$\le \max_{\boldsymbol{\lambda}\in\Lambda} \min_{\mathbf{u}\in\mathcal{U}} g(\boldsymbol{\lambda}, \mathbf{u}). \tag{24}$$

Eq. (21) is because the $\mathbf{u}$-player is playing best response so that $\mathbf{u}_t = \text{argmin}_{\mathbf{u}\in\mathcal{U}} g(\boldsymbol{\lambda}_t, \mathbf{u})$. Eq. (22) is because taking the minimum of each term of a sum cannot exceed the minimum of the sum as a whole. Eqs. (23) and (24) use the concavity of $g$ with respect to $\boldsymbol{\lambda}$, and the definition of $\max$, respectively. By letting $\delta = \frac{1}{T}\text{Regret}_T$, writing the definition of regret for the $\boldsymbol{\lambda}$-player, and using $\ell_t(\boldsymbol{\lambda}) = -g(\boldsymbol{\lambda}, \mathbf{u}_t)$, we have

$$\frac{1}{T}\sum_{t=1}^{T} g(\boldsymbol{\lambda}_t, \mathbf{u}_t) + \delta = \frac{1}{T}\max_{\boldsymbol{\lambda}\in\Lambda}\sum_{t=1}^{T} g(\boldsymbol{\lambda}, \mathbf{u}_t) \ge \max_{\boldsymbol{\lambda}\in\Lambda} g\left(\boldsymbol{\lambda}, \frac{1}{T}\sum_{t=1}^{T}\mathbf{u}_t\right) \ge \min_{\mathbf{u}\in\mathcal{U}}\max_{\boldsymbol{\lambda}\in\Lambda} g(\boldsymbol{\lambda}, \mathbf{u}),$$

where the second and third inequalities use convexity of $g$ with respect to $\mathbf{u}$ and definition of $\min$, respectively. Combining yields

$$\min_{\mathbf{u}\in\mathcal{U}} g\left(\frac{1}{T}\sum_{t=1}^{T} \boldsymbol{\lambda}_t, \mathbf{u}\right) \ge \min_{\mathbf{u}\in\mathcal{U}}\max_{\boldsymbol{\lambda}\in\Lambda} g(\boldsymbol{\lambda}, \mathbf{u}) - \delta,$$

and also

$$\max_{\boldsymbol{\lambda}\in\Lambda} g\left(\boldsymbol{\lambda}, \frac{1}{T}\sum_{t=1}^{T}\mathbf{u}_t\right) \le \max_{\boldsymbol{\lambda}\in\Lambda}\min_{\mathbf{u}\in\mathcal{U}} g(\boldsymbol{\lambda}, \mathbf{u}) + \delta,$$

completing the proof.

# C   Proof of Theorem 3.3

Let $v$ be the value of the game in Eq. (7):

$$v = \min_{\mu \in \Delta(\Pi)} \text{dist}(\bar{\mathbf{z}}(\mu), \mathcal{C}), \tag{25}$$

and let $\ell_t(\boldsymbol{\lambda}) = -\boldsymbol{\lambda} \cdot \hat{\mathbf{z}}_t$ (i.e., the loss function that OGD observes).

**Lemma C.1.** *For $t = 1, 2, \ldots, T$ we have*

$$\ell_t(\boldsymbol{\lambda}_t) = -\boldsymbol{\lambda}_t \cdot \hat{\mathbf{z}}_t \geq -v - (\epsilon_0 + \epsilon_1).$$

*Proof.* By Eq. (5) (which must hold by Lemma 3.2), and by Eq. (25), there exists $\mu^* \in \Delta(\Pi)$ such that

$$v = \text{dist}(\bar{\mathbf{z}}(\mu^*), \mathcal{C}) = \max_{\boldsymbol{\lambda} \in \Lambda} \boldsymbol{\lambda} \cdot \bar{\mathbf{z}}(\mu^*).$$

Thus, $\boldsymbol{\lambda}_t \cdot \bar{\mathbf{z}}(\mu^*) \leq v$ since $\boldsymbol{\lambda}_t \in \Lambda$ for all $t$. By our assumed guarantee for the policy $\pi_t$ returned by the planning oracle, we have

$$-\boldsymbol{\lambda}_t \cdot \bar{\mathbf{z}}(\pi_t) \geq -\boldsymbol{\lambda}_t \cdot \bar{\mathbf{z}}(\mu^*) - \epsilon_0 \geq -v - \epsilon_0.$$

Now using the error bound of the estimation oracle,

$$\|\bar{\mathbf{z}}(\pi_t) - \hat{\mathbf{z}}_t\| \leq \epsilon_1, \tag{26}$$

and the fact that $\|\boldsymbol{\lambda}_t\| \leq 1$, we have

$$(-\boldsymbol{\lambda}_t \cdot \hat{\mathbf{z}}_t) + \epsilon_1 \geq -\boldsymbol{\lambda}_t \cdot \bar{\mathbf{z}}(\pi_t).$$

Combining completes the proof. $\qquad\square$

Now we are ready to prove Theorem 3.3. Using the definition of mixed policy $\bar{\mu}$ returned by the algorithm we have

$$
\begin{aligned}
\text{dist}(\bar{\mathbf{z}}(\bar{\mu}), \mathcal{C}) &= \text{dist}\left(\frac{1}{T} \sum_{t=1}^{T} \bar{\mathbf{z}}(\pi_t), \mathcal{C}\right) \\
&= \max_{\boldsymbol{\lambda} \in \Lambda} \boldsymbol{\lambda} \cdot \left(\frac{1}{T} \sum_{t=1}^{T} \bar{\mathbf{z}}(\pi_t)\right) \qquad (27) \\
&= \frac{1}{T} \max_{\boldsymbol{\lambda} \in \Lambda} \sum_{t=1}^{T} \boldsymbol{\lambda} \cdot \bar{\mathbf{z}}(\pi_t) \\
&\leq \frac{1}{T} \max_{\boldsymbol{\lambda} \in \Lambda} \sum_{t=1}^{T} \boldsymbol{\lambda} \cdot \hat{\mathbf{z}}_t + \epsilon_1 \qquad (28) \\
&= -\frac{1}{T} \min_{\boldsymbol{\lambda} \in \Lambda} \sum_{t=1}^{T} \ell_t(\boldsymbol{\lambda}) + \epsilon_1 \qquad (29) \\
&\leq -\frac{1}{T} \min_{\boldsymbol{\lambda} \in \Lambda} \sum_{t=1}^{T} \ell_t(\boldsymbol{\lambda}) + \epsilon_1 + \frac{1}{T} \sum_{t=1}^{T} (\ell_t(\boldsymbol{\lambda}_t) + \epsilon_1 + \epsilon_0 + v) \qquad (30) \\
&= v + \left(-\frac{1}{T} \min_{\boldsymbol{\lambda} \in \Lambda} \sum_{t=1}^{T} \ell_t(\boldsymbol{\lambda}) + \frac{1}{T} \sum_{t=1}^{T} \ell_t(\boldsymbol{\lambda}_t)\right) + 2\epsilon_1 + \epsilon_0 \\
&= v + \frac{\text{Regret}_T(\text{OGD})}{T} + 2\epsilon_1 + \epsilon_0.
\end{aligned}
$$

Here, Eq. (27) is by Eq. (5). Eq. (28) uses Eq. (26) and the fact that $\|\boldsymbol{\lambda}\| \leq 1$. Eq. (31) uses Lemma C.1.

The diameter of decision set $\Lambda = \mathcal{C}^\circ \cap \mathcal{B}$ is at most 1. The gradient of the loss function $\nabla(\ell_t(\boldsymbol{\lambda})) = -\hat{\mathbf{z}}_t$ has norm at most $\|\bar{\mathbf{z}}(\pi_t)\| + \epsilon_1 \leq \frac{B}{1-\gamma} + \epsilon_1$. Therefore, setting $\eta = \left(\left(\frac{B}{1-\gamma} + \epsilon_1\right)\sqrt{T}\right)^{-1}$ based on Theorem A.1, we get

$$\frac{\text{Regret}_T(\text{OGD})}{T} \leq \left(\frac{B}{1-\gamma} + \epsilon_1\right) T^{-1/2}$$

# D  APPROPO for feasibility

---

**Algorithm 4** APPROPO – Feasibility

---

1: **input** projection oracle $\Gamma_{\mathcal{C}}(\cdot)$ for target set $\mathcal{C}$ which is a convex cone,
         positive response oracle $\text{PosPlan}(\cdot)$, estimation oracle $\text{Est}(\cdot)$,
         step size $\eta$, number of iterations $T$
2: **define** $\Lambda \triangleq \mathcal{C}^{\circ} \cap \mathcal{B}$, and its projection operator $\Gamma_{\Lambda}(\mathbf{x}) \triangleq (\mathbf{x} - \Gamma_{\mathcal{C}}(\mathbf{x}))/\max\{1, \|\mathbf{x} - \Gamma_{\mathcal{C}}(\mathbf{x})\|\}$
3: **initialize** $\boldsymbol{\lambda}_1$ arbitrarily in $\Lambda$
4: **for** $t = 1$ **to** $T$ **do**
5:      Call positive response oracle for the standard RL with scalar reward $r = -\boldsymbol{\lambda}_t \cdot \mathbf{z}$:
         $\pi_t \leftarrow \text{PosPlan}(\boldsymbol{\lambda}_t)$
6:      Call the estimation oracle to approximate long-term measurement for $\pi_t$:
         $\hat{\mathbf{z}}_t \leftarrow \text{Est}(\pi_t)$
7:      Update using online gradient descent with the loss function $\ell_t(\boldsymbol{\lambda}) = -\boldsymbol{\lambda} \cdot \hat{\mathbf{z}}_t$:
         $\boldsymbol{\lambda}_{t+1} \leftarrow \Gamma_{\Lambda}\big(\boldsymbol{\lambda}_t + \eta \hat{\mathbf{z}}_t\big)$
8:      **if** $\ell_t(\boldsymbol{\lambda}_t) < -(\epsilon_0 + \epsilon_1)$ **then**
9:          **return** *problem is infeasible*
10:      **end if**
11: **end for**
12: **return** $\bar{\mu}$, a uniform mixture over $\pi_1, \ldots, \pi_T$

---

## D.1  Proof of Theorem 3.4

**Lemma D.1.** *If the problem is feasible, then for $t = 1, 2, \ldots, T$ we have*

$$\ell_t(\boldsymbol{\lambda}_t) = -\boldsymbol{\lambda}_t \cdot \hat{\mathbf{z}}_t \geq -(\epsilon_0 + \epsilon_1).$$

*Proof.* If the problem is feasible, then there exists $\mu^*$ such that $\overline{\mathbf{z}}(\mu^*) \in \mathcal{C}$. Since all $\boldsymbol{\lambda}_t \in \mathcal{C}^{\circ}$, they all have non-positive inner product with every point in $\mathcal{C}$ including $\overline{\mathbf{z}}(\mu^*)$. Since $-\boldsymbol{\lambda}_t \cdot \overline{\mathbf{z}}(\mu^*) \geq 0$, we can conclude that $\max_{\pi \in \Pi} R(\pi) = \max_{\pi \in \Pi} -\boldsymbol{\lambda}_t \cdot \overline{\mathbf{z}}(\pi) \geq 0$. Therefore, by our guarantee for the positive response oracle,

$$R(\pi_t) = -\boldsymbol{\lambda}_t \cdot \overline{\mathbf{z}}(\pi) \geq -\epsilon_0.$$

Now using Eq. (26) and the fact that $\|\boldsymbol{\lambda}_t\| \leq 1$, we have

$$(-\boldsymbol{\lambda}_t \cdot \hat{\mathbf{z}}_t) + \epsilon_1 \geq -\boldsymbol{\lambda}_t \cdot \overline{\mathbf{z}}(\pi_t).$$

Combining completes the proof.                          □ The proof of Theorem 3.4 is similar to that of Theorem 3.3. If the algorithm reports infeasibility then the problem is infeasible as a result of Lemma D.1. Otherwise, we have

$$\frac{1}{T} \sum_{t=1}^{T} (\ell_t(\boldsymbol{\lambda}_t) + \epsilon_1 + \epsilon_0) \geq 0,$$

which can be combined with Eq. (29) as before. Continuing this argument as before yields

$$\text{dist}(\overline{\mathbf{z}}(\mu), \mathcal{C}) \leq \left(\frac{B}{1 - \gamma} + \epsilon_1\right) T^{-1/2} + 2\epsilon_1 + \epsilon_0,$$

completing the proof.

# E  Proof of Lemma 3.5

Let $\mathcal{C}' = \mathcal{C} \times \{\kappa\}$ and $\mathbf{q}$ be the projection of $\tilde{\mathbf{x}} = \mathbf{x} \oplus \langle \kappa \rangle$ onto $\tilde{\mathcal{C}} = \text{cone}(\mathcal{C}')$, i.e.,

$$\mathbf{q} = \arg\min_{\mathbf{y} \in \tilde{\mathcal{C}}} \|\tilde{\mathbf{x}} - \mathbf{y}\|.$$

Let $r$ be the last coordinate of $\mathbf{q}$. We prove the lemma in cases based on the value of $r$ (which cannot be negative by construction).

(a) $r > \kappa$

(b) $0 < r < \kappa$

(c) $r = 0$

Figure 3: Geometric Interpretation of the proof of Lemma 3.5

**Case 1** $(r > \kappa)$: Since $\mathbf{q} \in \mathrm{cone}(\mathcal{C}')$ with $r > 0$, there exists $\alpha > 0$ and $\mathbf{q}' \in \mathcal{C}'$ so that $\mathbf{q} = \alpha \mathbf{q}'$. See Figure 3a. Consider the plane defined by the three points $\tilde{\mathbf{x}}, \mathbf{q}, \mathbf{q}'$. Since the origin $\mathbf{0}$ is on the line passing through $\mathbf{q}$ and $\mathbf{q}'$, it must also be in this plane. Now consider the line that passes through $\tilde{\mathbf{x}}$ and $\mathbf{q}'$. Note that all points on this line have last coordinate equal to $\kappa$, and they are all also in the aforementioned plane. Let $\mathbf{v} \oplus \langle \kappa \rangle$ be the projection of $\mathbf{0}$ onto this line ($\mathbf{v} \in \mathbb{R}^d$).

Note that the two triangles $\Delta(\tilde{\mathbf{x}}, \mathbf{q}, \mathbf{q}')$ and $\Delta(\mathbf{0}, \mathbf{v} \oplus \langle \kappa \rangle, \mathbf{q}')$ are similar since they are right triangles with opposite angles at $\mathbf{q}'$. Therefore, by triangle similarity,

$$\frac{\|\mathbf{q}'\|}{\|\mathbf{v} \oplus \langle \kappa \rangle\|} = \frac{\|\tilde{\mathbf{x}} - \mathbf{q}'\|}{\|\tilde{\mathbf{x}} - \mathbf{q}\|} \geq \frac{\text{dist}(\tilde{\mathbf{x}}, \mathcal{C}')}{\text{dist}(\tilde{\mathbf{x}}, \tilde{\mathcal{C}})} = \frac{\text{dist}(\mathbf{x}, \mathcal{C})}{\text{dist}(\tilde{\mathbf{x}}, \tilde{\mathcal{C}})}.$$

Since $\mathbf{q}' \in \mathcal{C}'$, we have $\|\mathbf{q}'\| \leq \sqrt{(\max_{\mathbf{x} \in \mathcal{C}} \|\mathbf{x}\|)^2 + \kappa^2}$, resulting in

$$\frac{\|\mathbf{q}'\|}{\|\mathbf{v} \oplus \langle \kappa \rangle\|} \leq \frac{\sqrt{(\max_{\mathbf{x} \in \mathcal{C}} \|\mathbf{x}\|)^2 + \kappa^2}}{\kappa} = \sqrt{1 + 2\delta} \leq 1 + \delta$$

by the choice of $\kappa$ given in the lemma. Combining completes the proof for this case.

**Case 2 ($r = \kappa$):** Since $\mathbf{q} \in \text{cone}(\mathcal{C}')$ with $\kappa$ as last coordinate, we have $\mathbf{q} \in \mathcal{C}'$. Thus,

$$\text{dist}(\mathbf{x}, \mathcal{C}) = \text{dist}(\tilde{\mathbf{x}}, \mathcal{C}') \leq \|\tilde{\mathbf{x}} - \mathbf{q}\| = \text{dist}(\tilde{\mathbf{x}}, \tilde{\mathcal{C}})$$

which completes the proof for this case.

**Case 3 ($0 < r < \kappa$):** The proof for this case is formally identical to that of Case 1, except that, in this case, the two triangles $\Delta(\tilde{\mathbf{x}}, \mathbf{q}, \mathbf{q}')$ and $\Delta(\mathbf{0}, \mathbf{v} \oplus \langle \kappa \rangle, \mathbf{q}')$ are now similar as a result of being right triangles with a shared angle at $\mathbf{q}'$. See Figure 3b.

**Case 4 ($r = 0$):** Since $\mathbf{q} \in \text{cone}(\mathcal{C}')$, $\mathbf{q}$ must have been generated by multiplying some $\alpha \geq 0$ by some point in $\mathcal{C}'$. Since all points in $\mathcal{C}'$ have last coordinate equal to $\kappa > 0$, and since $r = 0$, it must be the case that $\alpha = 0$, and thus, $\mathbf{q} = \mathbf{0}$. Let $\mathbf{q}'$ be the projection of $\tilde{\mathbf{x}}$ onto $\mathcal{C}'$. See Figure 3c. Consider the plane defined by the three points $\tilde{\mathbf{x}}, \mathbf{q} = \mathbf{0}, \mathbf{q}'$. Let $\mathbf{q}''$ be the projection of $\tilde{\mathbf{x}}$ onto the line passing through $\mathbf{q}$ and $\mathbf{q}'$. Then

$$\|\tilde{\mathbf{x}} - \mathbf{q}''\| \leq \|\tilde{\mathbf{x}}\| = \text{dist}(\tilde{\mathbf{x}}, \tilde{\mathcal{C}}).$$

Now consider the line passing through $\tilde{\mathbf{x}}$ and $\mathbf{q}'$. Note that all points on this line have last coordinate equal to $\kappa$ and are also in the aforementioned plane. Let $\mathbf{v} \oplus \langle \kappa \rangle$ be the projection of $\mathbf{0}$ onto this line ($\mathbf{v} \in \mathbb{R}^d$). Note that the two triangles $\Delta(\tilde{\mathbf{x}}, \mathbf{q}'', \mathbf{q}')$ and $\Delta(\mathbf{0}, \mathbf{v} \oplus \langle \kappa \rangle, \mathbf{q}')$ are similar since they are right triangles with a shared angle at $\mathbf{q}'$. Therefore, by triangle similarity,

$$\frac{\|\mathbf{q}'\|}{\|\mathbf{v} \oplus \langle \kappa \rangle\|} = \frac{\|\tilde{\mathbf{x}} - \mathbf{q}'\|}{\|\tilde{\mathbf{x}} - \mathbf{q}''\|} \geq \frac{\text{dist}(\tilde{\mathbf{x}}, \mathcal{C}')}{\text{dist}(\tilde{\mathbf{x}}, \tilde{\mathcal{C}})} = \frac{\text{dist}(\mathbf{x}, \mathcal{C})}{\text{dist}(\tilde{\mathbf{x}}, \tilde{\mathcal{C}})}.$$

The rest of the proof for this case is exactly as in Case 1.

## F   Additional experimental details

All the models were trained using the following hyperparameters: policy network consists of 2-layer fully-connected MLP with ReLU activation and 128 hidden units and a A2C learning rate of $10^{-2}$. For APPROPO, the constant $\kappa$ (§3.3) is set to be 20. In the following figures, the performance of the algorithms has been depicted using different hyperparamters; showing average and standard deviation over 25 runs,.

(a) $n = 10$

(b) $n = 20$

Figure 4: Performance of APPROPO using different hyperparameters. The two numbers are learning rate for the online learning algorithm and $n$ (§3.4) respectively. In all figures, the x-axis is number samples. The vertical axes correspond to the three constraints, with thresholds shown as a dashed line; for reward (middle) this is a lower bound; for the others it is an upper bound.

(a) $n = 10$

(b) $n = 20$

Figure 5: Performance of APPROPO with diversity constraints using different hyperparameters. The two numbers are learning rate for the online learning algorithm and $n$ (§3.4) respectively. In all figures, the x-axis is number samples. The vertical axes correspond to the three constraints, with thresholds shown as a dashed line; for reward (middle) this is a lower bound; for the others it is an upper bound.

Figure 6: Performance of RCPO using different learning rates for Lagrange multiplier. In all figures, the x-axis is number samples. The vertical axes correspond to the three constraints, with thresholds shown as a dashed line; for reward (middle) this is a lower bound; for the others it is an upper bound.