[Reviews · NeurIPS 2019]

Reviewer 1



The paper presents a way to solve the approachibility problem in RL by reduction to a standard RL problem. It casts this problem as a zero-sum game using conic duality, which is solved by a primal-dual technique based on tools from online learning. The proposed algorithm assumes an oracle that approximately solves a standard RL problem. It runs primal-dual iterations, where the dual part of the algorithm updates measurement weights according to the current primal solution obtained from the oracle. Originality: This work introduces a new problem of finding policy those measurements vectors lies inside a convex target set. The presented approach largely builds on the ideas from Abernethy et al. The high-level algorithm (Algorithm 1) and the performance bound (Theorem 3.1) are known in the literature and are similar to those of Le et al. (Algorithm 1, Proposition 3.1) with the conic duality being used instead of Lagrangian duality. The proposed algorithm is novel to the RL community. The theoretical performance guarantee is established (Theorem 3.3) that relates the approximation errors at its subroutines and the number of rounds to the performance gap. Quality: The paper is theoretically sound. It supports the proposed algorithm with performance guarantee and the proofs are clearly presented. However, I have doubts regarding the practical feasibility of the algorithm that involves approximately solving an RL problem at each iteration. Even though authors describe computational shortcuts in Section 3.4, the effective running time and required memory resources is not reported in the experiments. Related concern is about the comparison at a fixed the number of trajectories (Figure 1). If my understanding is correct, the proposed algorithm does much more computation at a fixed number of trajectories than RCPO that only does 1 gradient descent step at each transition. Commenting on experiments, line 284 states that the algorithm achieves a “larger reward faster by the baseline”, but it is unclear if it is due to higher complexity of ApproPO. Moreover, it was not the part of the objective. Clarity: The paper is well-written, but it might not be straightforward for a reader from RL community. The paper would benefit from better the link/integration with the RL literature; explanation of the policy space convexification as it is not standard; explicitly stating time/space complexity of the algorithm; explicitly stating the two problems studied: the feasibility problem and distance minimization in Section 2 and link them to relevant sections. Significance: This paper presents a novel approachibility problem that might be of interest to the RL practitioners. The paper introduces to the RL community an approach from Abernethy et al. that combines ideas of conic duality and online learning. Although, the practical feasibility of this algorithm remains a concern, it might encourage other researchers to pursue this direction.

Reviewer 2



As far as I am aware, the contribution of this paper is novel. The authors propose a novel method for dealing with constraints in RL by solving a minimax game. The paper is very well written with great attention to detail, and I believe this could be of interest to the RL community given the increasing interest in safety. There are a few details that were not completely clear, so I would appreciate the effort in trying to address my concerns: - line 79: what is Delta(pi)? I don't think this was properly introduced. - line 90: if z is a vector and C a set, how do you define a distance between them? Is is the distance between z and the projected point to C? If so, what is the impact on the type of projection being used (e.g., orthogonal, oblique)? - Are lambda and zeta vectors? Is the reward represented as the dot product of these vectors? If so, the reward is a still a scalar, so how is this different from specifying the constraints as an added term in the reward function of standard RL? This is not completely clear to me. - I would appreciate some more general comment on the formulation. How does this game-theoretic framework compare to other techniques like projected methods? For ex. PNAC [1]. Why should one choose a game-theoretic approach over projected methods? Granted, I am not very familiar with the game-theory literature, but it is not clear to me what the benefit would be. Could you elaborate on that? - Figure 1: why are results a function of trajectories? Is this the same as episodes? [1] Thomas P., Dabney W., Giguere S., Mahadevan S. Projected Natural Actor Critic (NIPS 2013)

Reviewer 3



After reading all the reviews and the rebuttal, I still think that it is a good paper. The authors have also addressed my concern about the running time of the algorithm. The paper considers reinforcement learning problems where the goal is to find a policy such that an expected measurement vector satisfies some fixed convex constraints. In the RL setting, this problem can be seen as an approachability problem. Therefore, after formulating it as a two-player game, the authors propose to solve it with a no-regret learning algorithm. Regrets bound are also derived from existing ones in online convex optimization. In the reproducibility checklist, the authors state that a link to a downloadable source code is provided, however I have not found any in the paper or the supplemental material. * Originality Recently several works have considered constrained RL problems. This paper tackles a more general framework than those previous works by accepting any convex constraints. The reformulation of the overall problem as a two-player game is interesting and relatively novel, although it is not surprising for someone familiar with online convex optimization (see Chap. 8 of "Introduction to Online Convex Optimization" by Hazan). I think the authors should compare their approach with the following work: @article{KalathilBorkarJain14, Author = {D. Kalathil and V.S. Borkar and R. Jain}, Journal = {Arxiv}, Title = {A Learning Scheme for {Blackwell's} Approachability in {MDPs} and {Stackelberg} Stochastic Games}, Year = {2014}} * Quality The theoretical results seem to be correct. The overall game-theoretic formulation and the proof techniques are based on results from online convex optimization (regret bounds) and from Abernethy et al. (distance as max of linear function, cone from convex set). It seems to me that some previous work (e.g., RCPO) on constrained reinforcement learning could have been easily extended to deal with convex constraints. It would have been nice to have some discussion about this. The experiments are a bit limited (only one small domain and one baseline). I think it would have been interesting to compare with other concurrent methods for constrained reinforcement learning (e.g., CPO or the work by Kalathil et al.). In Fig. 1, the plots for APPROPO end earlier than the other ones. I guess that the authors have not finished running them at the submission time. This is probably due to the fact that the proposed method is quite computationally heavy, I think some comments about the running times of APROPO and RCPO should be provided in Section 4. * Clarity The paper is well-written and clear. * Significance The advantages of the method is that it is a general scheme that can accept more general convex constraints and also it has some performance guarantees. The drawbacks of the approach is that the obtained solution is a mixed policy, which is not very convenient to apply in practice and also the computational costs of the method. In my opinion, these make APROPO more of a theoretical proposition, which may perhaps pave the way for more practical and efficient algorithms in the future. Some typos: - l.149: Z(\mu) is used for the first time here. Should it be \bar z(\mu)? - l.152: "This means ..." is only true if the problem is feasible? - l.155: must to implement - l.181: apprimately - l.260: we previous

[Author Response · NeurIPS 2019]

We thank the reviewers for their constructive comments. We address the main concerns below.

**R1/R3: Running time and practicality of ApproPO:** In our experiments, we implement an RL oracle by a policy-gradient algorithm similar to RCPO (see Sec. 4), so our oracle has a similar *per-transition* running time as RCPO. The trajectories executed by our oracle have similar average lengths ($\approx$21) as those of RCPO ($\approx$24), so our oracle also has a similar *per-trajectory* running time as RCPO. We report the performance of our algorithm in terms of the total number of trajectories across all the RL oracle calls so far, so the part of the time that our algorithm spends in the RL oracle calls is commensurate with the RCPO time at any given number of trajectories. Our algorithm additionally performs updates of $\lambda$, but these are orders of magnitude cheaper than the per-trajectory running time of the RL oracle / RCPO, because the dimension of $\lambda$ is either 2 or 66 (without or with diversity constraints, respectively), whereas the policies $\pi$ over which the oracle and RCPO optimize are two-layer networks described by 8,704 floating-point numbers.

In our implementation, it was crucial to use the improvements from Sec. 3.4. We ran the "positive response" version of ApproPO (Algorithm 5) for 2000 outer-loop iterations (i.e., 2000 updates of $\lambda$), but needed to make at most 61 RL oracle calls in any of our replicates; the remaining outer-loop iterations recovered a positive response from cache. The cache look-up is fast since it only involves performing an inner product of $\lambda$ with stored expected measurement vectors.

The main overhead of ApproPO compared with RCPO is not the running time, but the memory, because of the policies stored in cache. This is quite modest (unless policy networks are very large), since the cache only stores the results of RL oracle calls. Note that the policy mixture returned by ApproPO is just a weighted combination of the policies from cache.

We will add this discussion to the paper and also update plots, so they are in terms of transitions rather than trajectories.

**R2: Clarification questions:**

- **definition of $\Delta(\Pi)$:** It is the set of probability distributions over policies in $\Pi$. The elements of $\Delta$ are viewed as mixed policies (see paragraph starting at line 73).

- **distance between vector z and a set $\mathcal{C}$:** You are right: we mean the distance to the closest point in $\mathcal{C}$ under the Euclidean distance (see Eq. (3)). Using other distance measures is open for future work; we expect it will allow us to use other no-regret algorithms and possibly improve convergence (in some cases).

- **reward as a dot product of $\lambda$ and z:** We construct the reward as a dot product only when we invoke the RL oracle—this is by design, since the oracle is a standard RL algorithm that seeks to optimize a scalar reward. Our algorithm, however, solves the multi-dimensional feasibility (or distance minimization) problem, which prior work does not handle. Our algorithm accomplishes this by systematically updating $\lambda$, calling the RL oracle for different $\lambda$ vectors, and then combining the returned policies into a mixture.

- **trajectories = episodes**: Yes, trajectories coincide with episodes.

**R2: ApproPO vs projected gradient methods, e.g., PNAC:** Projected gradient methods implement constraints in the *policy parameter space*, our constraints are on the observable measurements coming from the environment. The measurements have clear behavioral interpretation, so arguably constraining the measurements is quite natural. On the other hand, it can be quite hard to design constraints on the policy parameters. For example, it is not clear how to cast constraints from our experiments in terms of the weights parameterizing our two-layer policy networks.

**R3: Reproducibility checklist:** You are right; we apologize for this oversight. We ran out of time to anonymize the code properly. We are planning to publish the code with the camera-ready version.

**R3: Extending RCPO to convex constraints:** This was actually our original motivation, but in the end, a systematic solution turned out to be non-trivial and led to this paper (we are not aware of any simple extension in the literature).

**R3: Comparison with previous work:** Kalathil et al. (2014) and CPO (Achiam el at., 2017) address more restricted settings. Kalathil et al. focus on long-term average measurements in irreducible MDPs, their algorithm is a variant of $Q$-learning, and they only prove asymptotic convergence. CPO only handles orthant constraints, only supports discounted-sum rewards, and requires all iterates to satisfy the constraints (this is desirable in safe RL, but may be restrictive otherwise). Our approach works with both $Q$-learning and policy search oracles, supports both average measurements and discounted sums, and comes with non-asymptotic convergence guarantees. The setting of RCPO is similar to ours and it has demonstrated competitive empirical performance, so it was a natural baseline to include.

**R3: ApproPO plots end before RCPO plots:** This is because ApproPO terminates after 2000 outer loop iterations (note that the constraints are at that point approximately satisfied).

[Meta-Review · NeurIPS 2019]

The paper describes a new technique for RL with convex constraints. This is an important topic for robustness. The proposed technique is novel and significant. However, the experiments are somewhat preliminary. Nevertheless the paper makes an important contribution and it is clearly above the bar for publishing.